# Receptive field center-surround interactions mediate context-dependent spatial contrast encoding in the retina

Maxwell H Turner[1,2†*], Gregory W Schwartz[3,4], Fred Rieke[1]

[1]Department of Physiology and Biophysics, University of Washington, Seattle, United States; [2]Graduate Program in Neuroscience, University of Washington, Seattle, United States; [3]Departments of Ophthalmology and Physiology, Feinberg School of Medicine, Northwestern University, Chicago, United States; [4]Department of Neurobiology, Weinberg College of Arts and Sciences, Northwestern University, Chicago, United States

**Abstract** Antagonistic receptive field surrounds are a near-universal property of early sensory processing. A key assumption in many models for retinal ganglion cell encoding is that receptive field surrounds are added only to the fully formed center signal. But anatomical and functional observations indicate that surrounds are added before the summation of signals across receptive field subunits that creates the center. Here, we show that this receptive field architecture has an important consequence for spatial contrast encoding in the macaque monkey retina: the surround can control sensitivity to fine spatial structure by changing the way the center integrates visual information over space. The impact of the surround is particularly prominent when center and surround signals are correlated, as they are in natural stimuli. This effect of the surround differs substantially from classic center-surround models and raises the possibility that the surround plays unappreciated roles in shaping ganglion cell sensitivity to natural inputs.
DOI: https://doi.org/10.7554/eLife.38841.001

*For correspondence:
mhturner@stanford.edu

Present address: †Department of Neurobiology, Stanford University, Stanford, United States

## Introduction

The receptive field (RF) surround is a ubiquitous feature of early sensory computation. Surrounds in the retina have been proposed to enhance sensitivity to edges in a visual scene via lateral inhibition (*Marr and Hildreth, 1980*), and to promote efficient representation of naturalistic visual stimuli. Natural scenes contain strong correlations across visual space (*Dong and Atick, 1995*; *Field, 1987*; *Simoncelli and Olshausen, 2001*), and the RF surround can decorrelate responses across a population of visual neurons (*Atick, 2011*; *Atick and Redlich, 1992*; *Dan et al., 1996*), thereby increasing encoding efficiency by reducing redundancy in the population code (*Barlow, 2012*; *Srinivasan et al., 1982*). Recent findings have challenged this proposed function of the RF surround by showing that mechanisms other than the surround can play a larger role in decorrelating population responses to natural images. These include nonlinear processing in the retina (*Pitkow and Meister, 2012*) (see also [*Franke et al., 2017*; *Maheswaranathan et al., 2017*]) and fixational eye movements (*Boi et al., 2017*; *Kuang et al., 2012*; *Rucci and Victor, 2015*; *Segal et al., 2015*). These findings suggest a need to re-examine the role of the surround, especially as it relates to the encoding of natural visual stimuli.

Here, we answer several questions about center-surround RF structure and its relation to natural scenes. First, how does the circuit location of surround suppression (before or after summation over visual space) impact RGC computation? Second, does surround activation change the linearity of

spatial integration by RGCs? And finally, how do the statistics of natural scenes influence nonlinear interactions between the center and the surround?

Classical Difference-of-Gaussians (*Enroth-Cugell and Lennie, 1975*; *Enroth-Cugell and Robson, 1966*; *Rodieck and Stone, 1965*; *Rodieck, 1965*) and modern predictive RF models (*Heitman et al., 2016*; *Pillow et al., 2008*) assume that the surround interacts only with the fully formed center signal (*Figure 1A*). These models suggest that the surround modulates the gain of signals in the center but does not alter how the center integrates signals across space. However, surrounds are generated by horizontal cells in the outer retina (*Crook et al., 2011*; *Davenport et al., 2008*; *Mangel, 1991*; *McMahon et al., 2004*; *Verweij et al., 2003*; *Ströh et al., 2018*; *Drinnenberg et al., 2018*) and by amacrine cell feedback to bipolar cells in the inner retina (*Cook and McReynolds, 1998*; *Farrow et al., 2013*; *Flores-Herr et al., 2001*; *Hoggarth et al., 2015*; *Taylor, 1999*) (*Figure 1C*). As a result, at least a portion of the surround is present in the bipolar cells presynaptic to a RGC (*Figure 1C*, *Figure 2—figure supplement 1*; [*Dacey et al., 2000*; *Werblin and Dowling, 1969*; *Zhang et al., 2009*; *Borghuis et al., 2013*; *Franke et al., 2017*; *Buldyrev and Taylor, 2013*]) and hence is present prior to the summation over space that forms the center. Subunits with antagonistic surrounds have been incorporated into models of the RGC RF

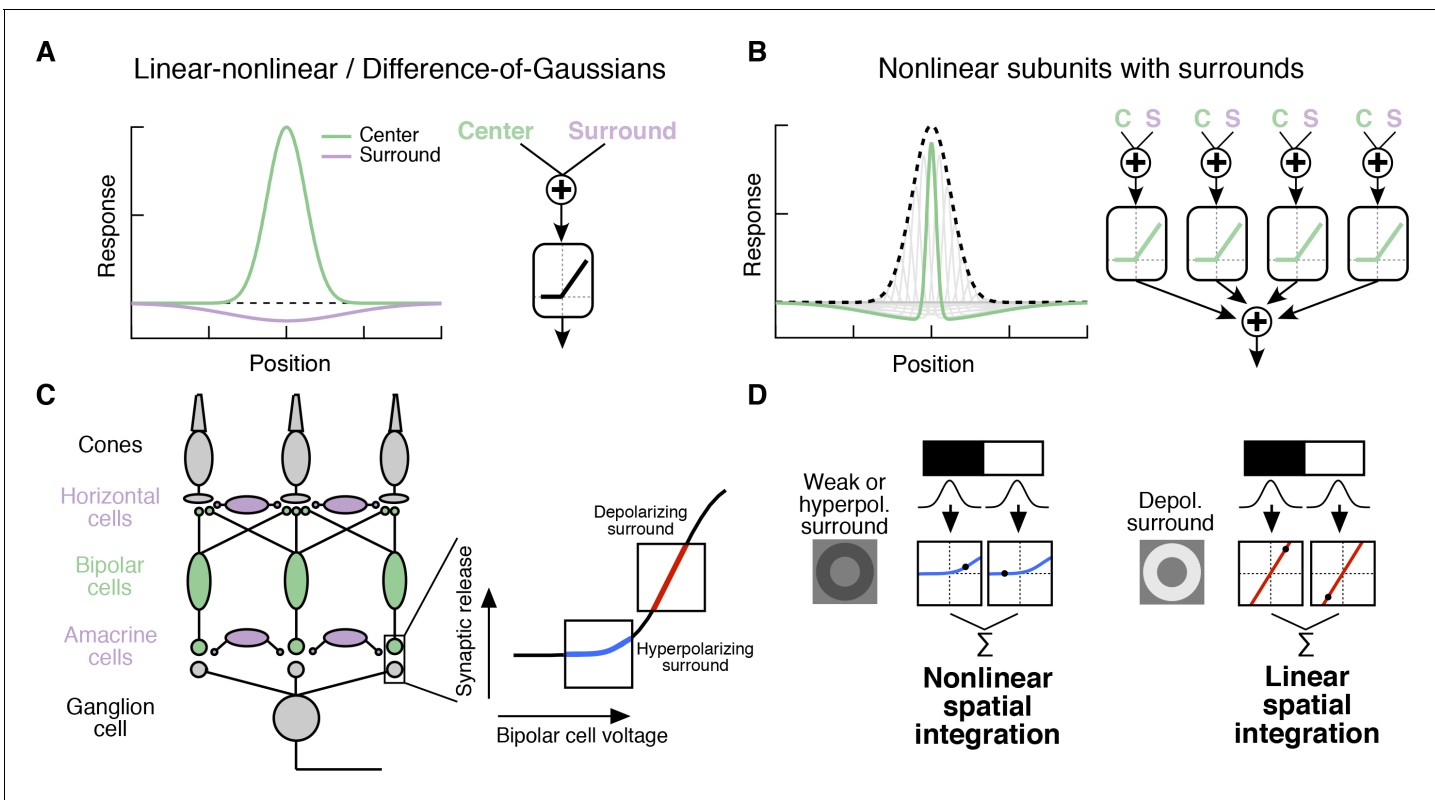

**Figure 1.** Circuit basis of the surround and implications for receptive field (RF) structure. (A,B) Two models for how the surround is integrated with the center to form the full RF: The linear-nonlinear model shown in (A) linearly combines the center and surround before treatment with an output nonlinearity that occurs after spatial integration. The model in (B) combines center and surround signals within each nonlinear subunit, and treats the combined center-surround signal with a private nonlinearity. In other words, individual subunits in (B) have their own RF surrounds. (C) Retinal circuit schematic. Bipolar cells make excitatory synapses on to retinal ganglion cells (RGCs). The nonlinear synaptic transfer from bipolar cell to RGC means that bipolar cells act as nonlinear subunits within the RF center of the RGC. The surround is generated by horizontal and/or amacrine cells, both of which can influence bipolar cell responses upstream of the synaptic nonlinearity. Surrounds that hyperpolarize bipolar cells shift the synapse into a more rectified state (blue portion of synaptic nonlinearity). Surrounds that depolarize bipolar cells shift the synapse into a locally linear state (red portion of curve). (D) This circuit organization suggested a working model of how inputs to the surround could change RF structure in the center. Shown is a schematic illustrating this hypothesis for an Off-center RGC. A weak or hyperpolarizing surround will be associated with rectified subunit output nonlinearities, and thus nonlinear spatial integration and sensitivity to spatial contrast (e.g. a split-field grating stimulus). A depolarizing surround will be associated with more linear subunit outputs and the RGC should integrate across visual space approximately linearly.
DOI: https://doi.org/10.7554/eLife.38841.002

(*Enroth-Cugell and Freeman, 1987*). Nonetheless, the RF structure in *Figure 1B* is not reflected in modern, predictive RGC models nor have its functional consequences been explored.

For many RGC types bipolar cells act as excitatory, nonlinear subunits within the RF center (*Schwartz et al., 2012*; *Demb et al., 2001*) (*Figure 1C*). The degree of rectification at the bipolar cell to RGC synapse can control the spatial integration properties of the RF center (*Grimes et al., 2014*; *Turner and Rieke, 2016*). In particular, a weakly rectified or linear synapse yields linear spatial integration due to cancellation of bright and dark regions of a scene, whereas a sharply rectified bipolar output synapse yields nonlinear spatial integration because light and dark regions fail to cancel. Nonlinear spatial integration causes RGCs to respond strongly to stimuli with fine spatial structure like gratings (*Cafaro and Rieke, 2013*; *Crook et al., 2008*; *Enroth-Cugell and Robson, 1966*; *Hochstein and Shapley, 1976*; *Petrusca et al., 2007*) or natural images (*Turner and Rieke, 2016*).

We hypothesized that surround modulation could dynamically regulate nonlinear spatial integration in a RGC by controlling the effective rectification of the bipolar synapse. Thus, in response to weak surround input or surround input that hyperpolarizes bipolar cells, bipolar cell terminals presynaptic to the RGC are in a rectified state and the RF center integrates nonlinearly over visual space (*Figure 1C,D*). When surround input depolarizes bipolar cells in the RF center, however, the bipolar cell synapse is relatively more linear, and the RF center integrates linearly over visual space (*Figure 1C,D*). We test this hypothesis using both synthetic and natural visual stimuli to probe responses of parasol (magnocellular-projecting) RGCs in the macaque monkey retina. The nonlinear subunit structure of these cells has been well characterized (*Cafaro and Rieke, 2013*; *Crook et al., 2008*; *Petrusca et al., 2007*). We focus on Off-center parasol RGCs in particular because the nonlinear structure of their RF center is important for encoding natural visual stimuli (*Turner and Rieke, 2016*). Many of the features of center-surround interactions we explore here also apply to On parasol RGCs as well, suggesting that these interactions are general features of retinal computation. Our results show that RGC sensitivity to spatial contrast in natural scenes is modulated by context via a nonlinear interaction between the RF center and surround.

## Results

We used single-cell patch electrophysiology in an *in vitro* macaque retinal preparation in conjunction with computational modeling of the RGC RF to explore the impact of the surround on nonlinear RF structure and natural scene encoding. We started by testing the hypothesis outlined in *Figure 1C,D*. Consistent with this hypothesis, we found that the linearity of spatial integration in the RF center depends on surround activation. Next, we used natural and artificial stimuli to characterize nonlinear interactions between center and surround and to test circuit models for the origin of these interactions. Finally, we show that the intensity correlations characteristic of natural scenes promote nonlinear interactions and make spatial integration relatively insensitive to changes in local luminance across a visual scene.

### The RF surround regulates nonlinear spatial integration in the RF center

To test the hypothesis of *Figure 1C,D*, we systematically manipulated surround signals while probing spatial integration in the RF center. We focused this test on Off parasol RGCs, because these cells show both stronger rectification of subunit output than On parasol cells (*Chichilnisky and Kalmar, 2002*) and nonlinear spatial integration in the context of naturalistic visual stimuli (*Turner and Rieke, 2016*). We began each experiment by centering the stimulus over the RF and measuring the linear RF (see Materials and methods and *Figure 2—figure supplement 1*). We then tailored visual stimuli for each cell such that the 'center region' stimulus did not extend into the pure surround RF subregion and the 'surround region' stimulus did not cover the center. These subregions are not exclusively associated with a center or surround mechanism since the antagonistic surround is spatially coextensive with the RF center (*Figure 1A,B*). Nonetheless, according to estimated RFs, the 'center region' stimulus activated the center mechanism ~4 times more strongly than the surround mechanism and the 'surround region' stimulus activated the surround mechanism ~5 times more strongly than the center mechanism. This degree of specificity allowed us to ask questions about interactions between these two RF subregions.

We previously found that nonlinear spatial integration endows Off parasol RGCs with sensitivity to spatial contrast in natural images (*Turner and Rieke, 2016*). To test whether surround activity

modulates this spatial contrast sensitivity, we measured Off parasol RGC spike responses to natural image patches that contained high spatial contrast and were expected to activate the nonlinear component of the cell's response (see Materials and methods for details on images and patch selection). For each image patch, we also presented a linear equivalent disc stimulus, which is a uniform disc with intensity equal to a weighted sum of the pixel intensities within the RF center. The weighting function was an estimate of each cell's linear RF center from responses to expanding spots (see Materials and methods and *Figure 2—figure supplement 1*). A cell whose RF center behaves according to this linear RF model will respond equally to a natural image and its associated linear equivalent disc.

As shown previously, Off parasol cells responded much more strongly to natural images than to linear equivalent stimuli, especially when the natural image contains high spatial contrast (*Figure 2A*, left, *Figure 2B*; see also [*Turner and Rieke, 2016*]). However, when a bright surround was presented with the center stimulus, the natural image and its linear equivalent disc produced near-equal responses (*Figure 2A,B*). Dark surrounds strongly suppressed both responses (*Figure 2A*, right). If responses to center and surround stimuli added linearly, the difference between responses to the natural image and linear equivalent disc should be maintained across surrounds and the points in *Figure 2B* should lie on a line offset from the diagonal. This was clearly not the case.

We used the difference in spike count between a natural image and its linear equivalent disc as a metric of spatial contrast sensitivity. This difference, as shown in *Figure 2C*, depended systematically on the difference in mean intensity in the center and surround regions. Specifically, spatial contrast sensitivity was maximal in response to stimuli for which center and surround intensities were similar and dropped as intensity in these regions diverged. Hence, nonlinear spatial integration is maximized when center and surround experience similar mean luminance. When the surround strongly hyperpolarizes presynaptic bipolar cells (for Off cells, a dark surround), the response is diminished as a result of the surround shifting the synapse into a quiescent state. When the surround depolarizes presynaptic bipolar cells (for Off cells, a bright surround), the nonlinear sensitivity of the center is

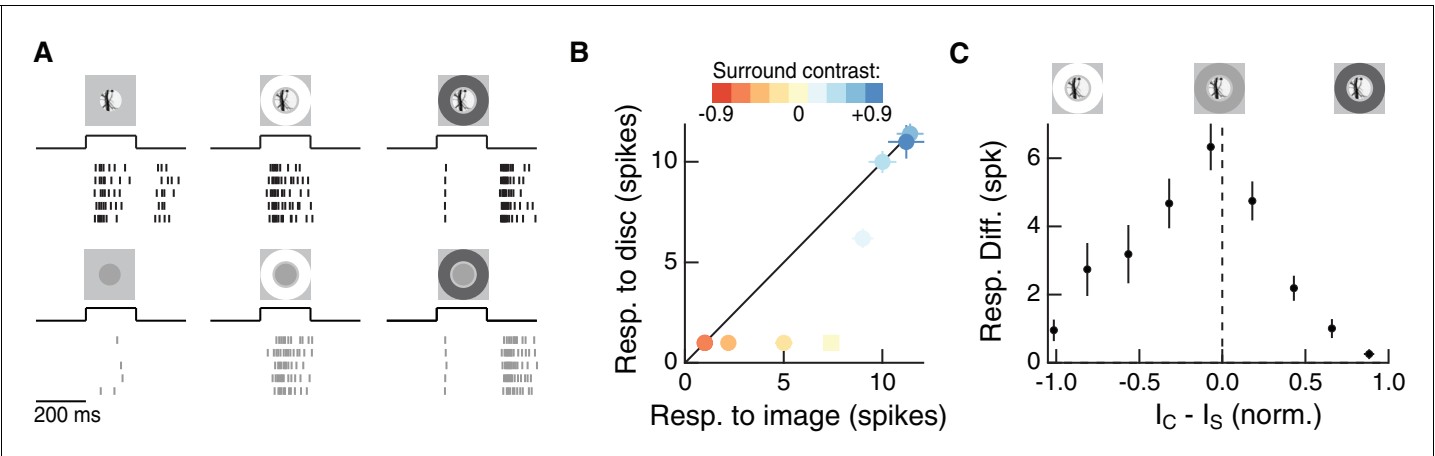

**Figure 2.** The RF surround regulates nonlinear spatial integration of natural images. (**A**) We presented a natural image patch and its linear-equivalent disc stimulus to probe sensitivity to spatial contrast in natural scenes. Rows of each raster correspond to repeated presentations of the same stimulus for the example Off parasol RGC. (**B**) Spike count responses to an example image patch and its linear equivalent disc across a range of surround contrasts. The addition of a sufficiently bright surround (top three points) eliminates sensitivity to spatial contrast in this image patch. (**C**) Population summary showing the response difference between image and disc as a function of the difference in mean intensity between the RF center and surround. Negative values of this difference correspond to a surround that is brighter than the center, and positive values to a surround that is darker than the center (n = 21 image patch responses measured in five Off parasol RGCs).

DOI: https://doi.org/10.7554/eLife.38841.003

The following source data and figure supplement are available for figure 2:

**Source data 1.** Included is a .mat file containing a data structure for the data in *Figure 2*.
DOI: https://doi.org/10.7554/eLife.38841.005
**Figure supplement 1.** Measuring linear center-surround structure in parasol RGCs.
DOI: https://doi.org/10.7554/eLife.38841.004

reduced. Hence, as in the hypothesis of *Figure 1*, these experiments indicate that the activity of the RF surround, rather than only interacting with the fully formed RF center signal, can control spatial integration by the center.

Does the impact of the surround on spatial integration in the center depend on specific statistics of natural images or is it a more general phenomenon? To answer this question, we repeated these experiments using a flashed split-field grating rather than natural image patches (*Figure 3*). Because of the nonlinear subunit structure of the RF center, Off parasol cells respond strongly to such stimuli (*Figure 3A*). We presented the same grating together with a surround annulus and, in separate trials, the surround annulus alone (*Figure 3A*). When paired with a bright surround, the grating stimulus did not activate the cell much beyond its response to the surround alone. A dark surround suppressed responses with and without the grating, save for a small, brief response at the beginning of the presentation of the grating which is likely the result of the brief temporal delay of the surround relative to the center. A similar early response was present for natural images with dark surrounds (*Figure 2A*).

We repeated this experiment for surround contrasts ranging from +0.9 to −0.9 (*Figure 3B*). While the surround-free grating (surround contrast = 0) stimulus showed strong nonlinear integration (indicated by its distance away from the unity line in *Figure 3B*), the presence of a surround stimulus diminished the response to the grating (indicated by the tendency of non-zero surround contrast points to lie closer to the line of unity). This was true for a range of central grating contrasts (*Figure 3C*), which indicates that this behavior is not the result of response saturation. Thus, just as for natural image patches, nonlinear spatial integration is maximal when the center and surround experience the same mean luminance (in this case a mean of zero) and decreases when the surround is brighter or dimmer than the center.

To test whether the effect of the surround on spatial integration was present in the bipolar synaptic output, we repeated these experiments while measuring a ganglion cell's excitatory synaptic inputs (*Figure 3D*; see Materials and methods for isolation of excitatory inputs). Modulation of spatial integration by surround activity was similar in excitatory inputs and spike responses (*Figure 3E, F*), which indicates that it is already present in the bipolar synaptic output and is not substantially shaped by post-synaptic integration or spike generation mechanisms. A similar effect can be seen in the excitatory inputs to On parasol RGCs (*Figure 3—figure supplement 1*). This shows that this effect of the surround is not unique to the Off parasol excitatory pathway, but may be a more general feature of center-surround RF organization in the retina. Compared to Off parasol RGCs, On parasol RGCs are more easily shifted into a regime of linear spatial integration, presumably because of the shallower rectification of nonlinear subunits in the receptive field of On parasol cells (*Chichilnisky and Kalmar, 2002*; *Turner and Rieke, 2016*).

The experiments described in *Figures 2* and *3* show that inputs to the RF surround can influence how the RF center integrates signals across space, consistent with the hypothesis outlined in *Figure 1*. For both the spike output and excitatory synaptic input to an Off parasol RGC, the peak spatial nonlinearity was observed when center and surround experienced similar mean luminance (*Figure 2C* and *Figure 3C,F*).

## Nonlinear center-surround interactions are dominated by a single, shared nonlinearity

The hypothesis in *Figure 1* relies on a specific form of nonlinear center-surround interaction, whereby center and surround signals combine upstream of a shared, rectifying nonlinearity. To characterize center-surround interactions in a more complete and unbiased manner, we used Gaussian-distributed noise stimulation and a linear-nonlinear cascade modeling approach. We presented Gaussian-distributed random noise to the center region alone, surround region alone or both regions together while measuring excitatory synaptic inputs to On and Off parasol cells (*Figure 4A*). While thus far we have focused exclusively on Off parasol RGCs, the hypothesis in *Figure 1* should also apply to excitatory input to On parasol RGCs, and hence we performed these experiments on both parasol types. For this analysis, we estimated the excitatory conductance by dividing the measured excitatory currents by the driving force. We computed linear filters for each RF region using reverse correlation based on trials in which the center or surround was stimulated in isolation (*Figure 4A*, left and middle columns).

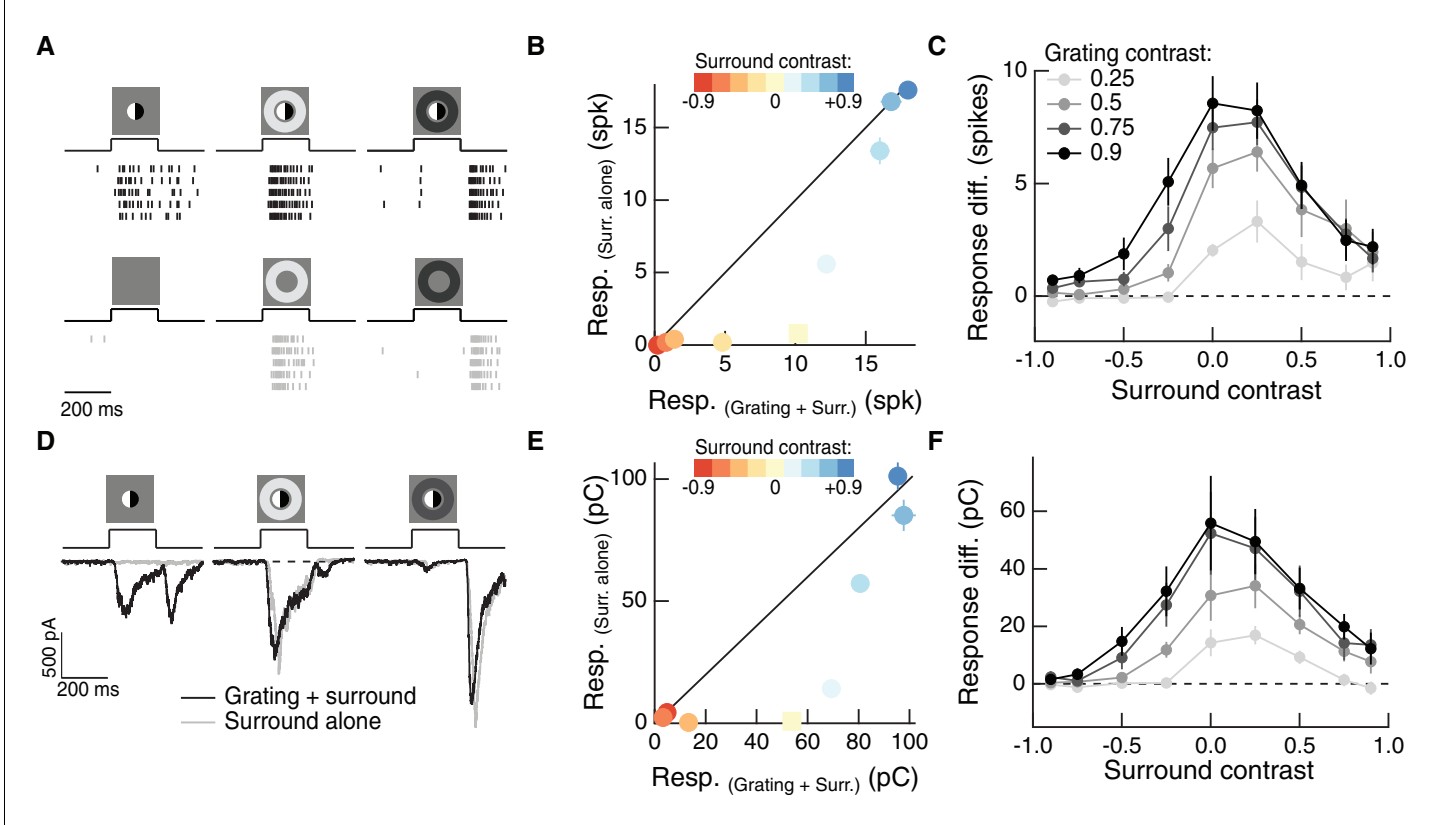

**Figure 3.** The RF surround regulates nonlinear spatial integration in the RF center. (**A**) Left column: Example Off parasol RGC spike response to an isolated split-field grating stimulus in the RF center. Rows of each raster correspond to repeated presentations of the same stimulus for the example cell. There is no linear equivalent stimulus in this case since the grating has a mean of zero. Center column: when the center stimulus is paired with a bright surround, the grating and the surround alone produce very similar spike responses. Right column: a dark surround suppresses the response in both cases, and the grating is unable to elicit a strong response. (**B**) For the example cell in (**A**), we tested sensitivity to the center grating stimulus with a range of contrasts presented to the surround. Negative contrast surrounds (hyperpolarizing for Off bipolar cells) decrease the response. Positive contrast surrounds (depolarizing for Off bipolar cells) sum sub-linearly with the grating stimulus such that for the brightest surrounds, the addition of the grating only mildly enhances the cell's response. Points show mean (± S.E.M.) spike count. (**C**) We measured the response difference between the grating stimulus and the surround-alone stimulus across a range of surround contrasts (horizontal axis) and for four different central grating contrasts (different lines). For each grating contrast, addition of either a bright or dark surround decreased sensitivity to the added grating. Points are population means (± S.E.M.) (n = 5 Off parasol RGCs). (**D–F**) same as (**A–C**) for excitatory synaptic current responses of an Off parasol RGC. Points represent mean (± S.E.M.) excitatory charge transfer for the example cell in (**E**) and population mean (±S.E.M.) (n = 7 Off parasol RGCs) in (**F**).

DOI: https://doi.org/10.7554/eLife.38841.006

The following source data and figure supplement are available for figure 3:

**Source data 1.** Included is a .mat file containing a data structure for the data in *Figure 3A–C*.
DOI: https://doi.org/10.7554/eLife.38841.008

**Source data 2.** Included is a .mat file containing a data structure for the data in *Figure 3D–F*.
DOI: https://doi.org/10.7554/eLife.38841.009

**Source data 3.** Included is a .mat file containing a data structure for the data in *Figure 3—figure supplement 1*.
DOI: https://doi.org/10.7554/eLife.38841.010

**Figure supplement 1.** Regulation of spatial integration by the RF surround of On parasol RGCs.
DOI: https://doi.org/10.7554/eLife.38841.007

We constructed three models of how center and surround inputs combine to determine the cell's excitatory conductance response when both RF regions are stimulated (see Materials and methods for details). (1) In the 'independent' model (*Figure 4B*), inputs to the center and surround are filtered using their respective linear filters and then passed through separate nonlinear functions. The outputs of the two nonlinearities are then summed to give the response of the cell. (2) In the 'shared' model (*Figure 4C*), filtered center and surround inputs are summed linearly before passing through

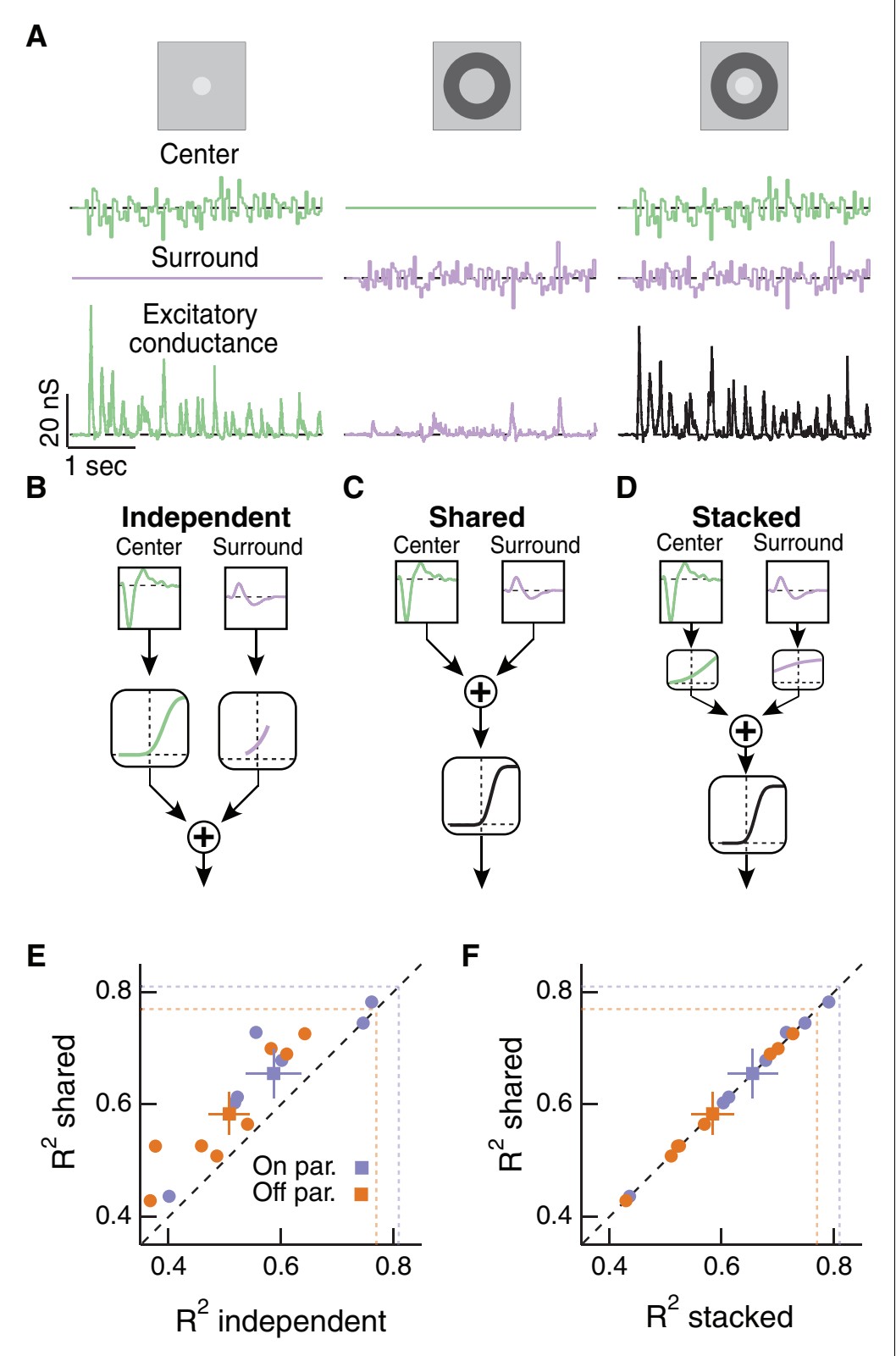

**Figure 4.** Linear-nonlinear cascade modeling supports an architecture where center and surround combine linearly before passing through a shared nonlinearity. (**A**) We presented Gaussian noise to either the center region (left), surround region (middle) or center and surround regions simultaneously (right) while measuring excitatory synaptic current responses. Measured excitatory currents (in pA) have been converted to excitatory conductance (in nS).

*Figure 4 continued on next page*

*Figure 4 continued*

Example traces are from a representative Off parasol RGC. (B) The independent model treats the filtered center and surround inputs with private nonlinear functions, and the outputs of these two nonlinearities are then summed to produce the excitatory conductance response. (C) The shared model integrates filtered center and surround inputs linearly, and this summed input is then passed through a single, shared nonlinearity. (D) The stacked model combines the independent and shared models by treating center and surround with private nonlinearities before summation and treatment with a third, shared nonlinearity. (E) We tested the ability of each of these models to predict held-out responses to center-surround stimulation. The shared model outperforms the independent model in both On and Off parasol RGCs (n = 7 On cells, $p = 0.03$; n = 8 Off cells, $p = 0.008$). (F) The fraction of explained variance was the same for the shared compared to the stacked model ($p>0.90$ for both On and Off cells). Dashed lines show estimates of the response reliability, which sets an upper bound for model performance (see Materials and methods for details).

DOI: https://doi.org/10.7554/eLife.38841.011
The following source data and figure supplement are available for figure 4:

**Source data 1.** Included is a .mat file containing a data structure for the data in *Figure 4 and 5*.
DOI: https://doi.org/10.7554/eLife.38841.013
**Figure supplement 1.** Representative predictions of linear-nonlinear cascade models for center-surround interactions.
DOI: https://doi.org/10.7554/eLife.38841.012

a single, shared nonlinear function. The output of this nonlinearity is the cell's response. (3) The 'stacked' model (*Figure 4D*) combines models (1) and (2) by treating center and surround with private nonlinearities before summation and treatment with a third, shared nonlinearity. Representative predictions and measured responses are shown in *Figure 4—figure supplement 1*. The shared model is a special case of the stacked model, where the upstream independent functions are linear. Similarly, the independent model is a special case of the stacked model, where the output function after summation is linear.

We fit the nonlinear functions in each model using a subset of simultaneous center-surround trials and used the remaining simultaneous trials to test how well each model could predict the cell's response. These models generally captured ~60% of the total response variance, and ~80% of the explainable variance (see Materials and methods). While the models appear to perform better for On compared to Off parasol RGCs (see *Figure 4E,F*), this difference is not statistically significant (p=0.19). The shared model outperformed the independent model (*Figure 4E*). The shared model performed as well as the more complicated stacked model (*Figure 4F*), despite the latter having many more free parameters (10 free parameters) than the shared model (five free parameters). In addition, the private nonlinearities fit in the stacked model tended to be quite shallow and much nearer to linear than the sharply rectified shared nonlinearity (*Figure 4D*). Hence the stacked model in practice effectively behaved like the shared model. The modeling result in *Figure 4F* supports the hypothesis that the dominant nonlinear interaction between center and surround is characterized by a shared nonlinearity, and that upstream of this nonlinearity center and surround interact approximately linearly.

The hypothesis in *Figure 1* suggests that the effective rectification experienced by each subunit will depend on surround activation. The experiments used for the modeling above allowed us to directly examine whether this is the case. To do this, we estimated center and surround activation by convolving center and surround filters with the appropriate stimuli. We then plotted the measured excitatory conductance against these estimates of center and surround activation; *Figure 5A* shows an example for the same Off parasol RGC as *Figure 4A–D*.

These joint response distributions show that the relationship between center activation and excitatory conductance depends on surround activation. When the surround is only weakly activated (near zero on 'surround' axis in *Figure 5A*), this nonlinear relationship is rectified (*Figure 5B*). Rectification persists when the surround hyperpolarizes presynaptic bipolar cells (negative on 'surround' axis in *Figure 5A*, blue trace in *Figure 5A,B*). But when the surround depolarizes bipolar cells (positive on 'surround' axis in *Figure 5A*), the relationship between center activation and excitatory conductance becomes more linear (i.e. less rectified; *Figure 5A,B*, red trace). We quantified this change in center rectification with surround activation using a rectification index (RI, see Materials and

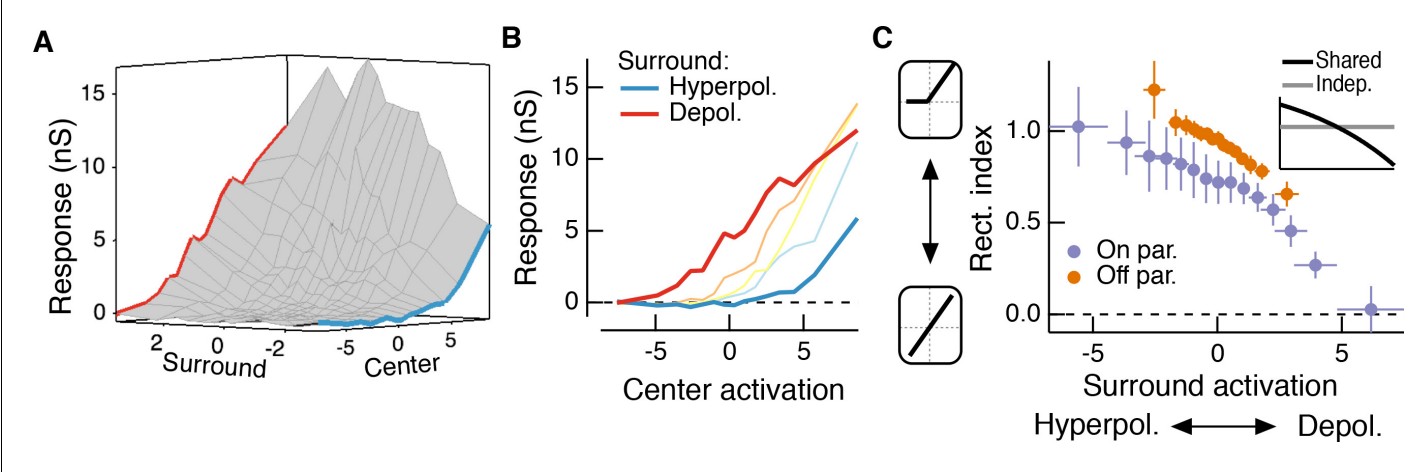

**Figure 5.** The RF surround changes the apparent rectification of inputs from the center. (A) Response surface showing the mean excitatory conductance response from an Off parasol RGC as a function of filtered inputs to both the center and surround (center or surround 'activation', that is their generator signals). (B) Sections through this surface at various levels of surround activation reveal that the shape of the nonlinear dependence of excitatory conductance on center activation changes as the surround is modulated. (C) To quantify this change in center rectification, we used a rectification index (see Materials and methods), where values near 0 indicate a linear relationship between center activation and conductance response, and values near one indicate a sharply rectified relationship. Points are mean (± S.E.M.) (n = 7 On parasol cells and n = 8 Off parasol cells). Inset shows the expected relationship between rectification index and surround activation for a shared nonlinearity model (black curve) and an independent nonlinearity model (gray curve).

DOI: https://doi.org/10.7554/eLife.38841.014

methods for calculation of this metric). A RI value of zero indicates a linear relationship between center activation and conductance response, whereas RI values near one indicate strong rectification (i.e. there is a large increase in response with positive center activation, but very little or no decrease in response with negative center activation). For both On and Off cells, rectification decreased as surround activation increased (*Figure 5C*). In agreement with previous observations (*Chichilnisky and Kalmar, 2002*; *Turner and Rieke, 2016*), Off cells were more rectified than On cells. The inset to *Figure 5C* shows the relationship between surround activation and RI for independent and shared nonlinearity models. When center and surround nonlinearities are independent, the rectification of the center does not depend on the activity of the surround because the surround enters only after the center is fully formed (horizontal gray line in inset). The shared nonlinearity model, on the other hand, predicts a decrease in rectification as the surround becomes more depolarizing, in agreement with the behavior of parasol RGCs.

The experiments described provide additional quantitative support for the circuit architecture of *Figure 1B* in which center and surround signals add linearly prior to a shared nonlinearity.

## RF center and surround interact nonlinearly during naturalistic visual stimulation

To test whether inputs to the RF center and surround interact nonlinearly under naturalistic stimulus conditions, we used a visual stimulus designed to approximate natural primate viewing conditions based on the Database Of Visual Eye movementS (DOVES, (*Van Der Linde et al., 2009*; *van Hateren and van der Schaaf, 1998*). An example image and a corresponding eye movement trajectory is shown in *Figure 6A*. We masked the stimulus to the RF center region, surround region or both. *Figure 6B* shows spike responses of an example Off parasol RGC to three movie stimuli: stimulation of the center region alone (*Figure 6B*, green), stimulation of the surround region alone (*Figure 6B*, purple), or simultaneous stimulation of both the center and surround regions (*Figure 6B*, black). Responses to isolated center or surround stimuli are shown in *Figure 6C* for average spike responses (*Figure 6C*, top) and excitatory synaptic inputs (*Figure 6C*, bottom).

To determine whether center and surround signals interact nonlinearly, we compared the linear sum of center and surround responses (*Figure 6D*, gray traces) to the measured response to

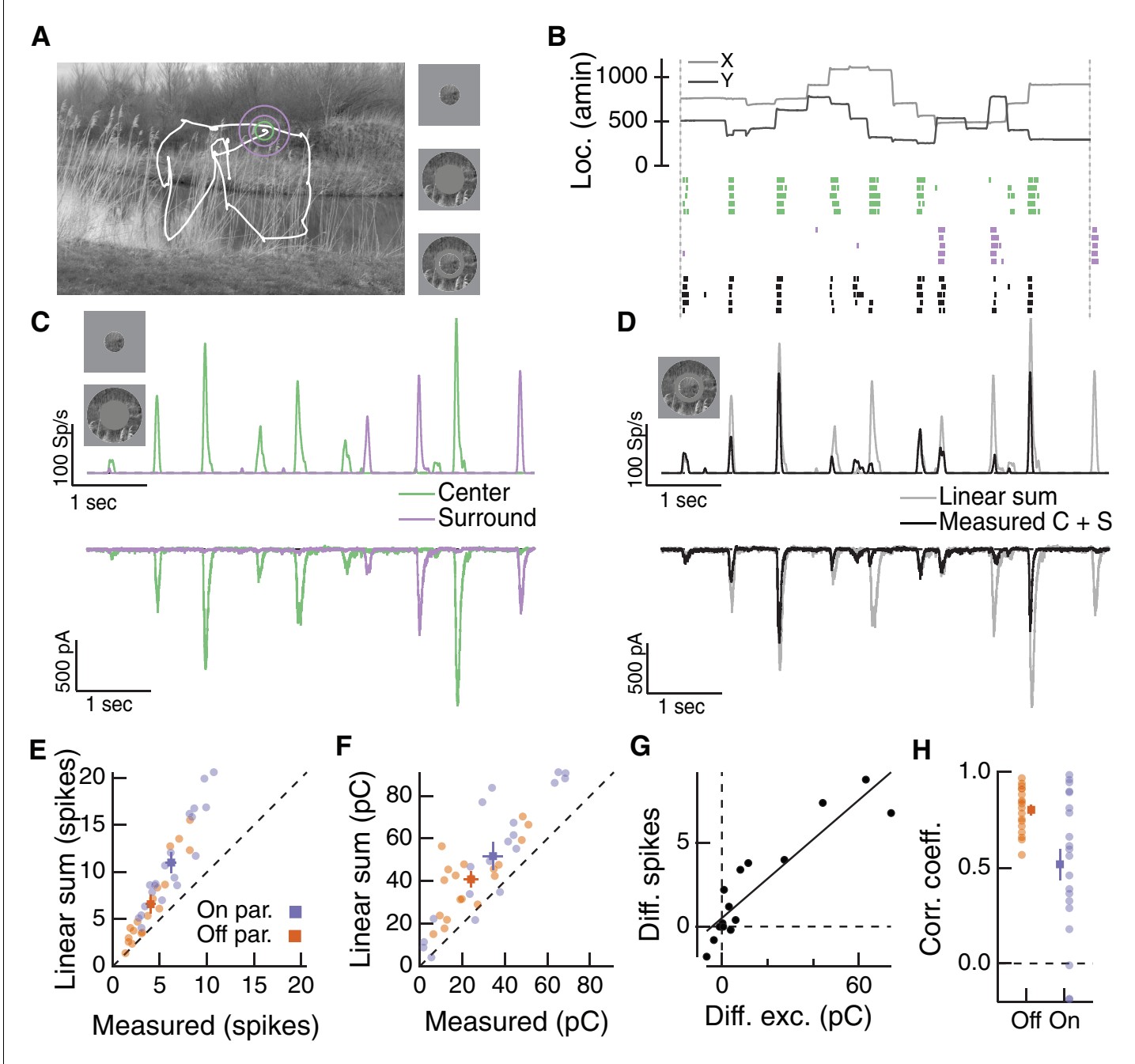

**Figure 6.** Natural movie stimuli elicit nonlinear interactions between the RF center and surround. (A) Natural image and associated eye movement trajectory from (*Van Der Linde et al., 2009*). Right: example movie frames showing isolated center (top), surround (middle), and center-surround stimuli (bottom). (B) Rasters show example Off parasol RGC spike responses to these three movie stimuli. Top shows eye movement position. (C) Spike output (top) and excitatory synaptic input (bottom) in response to isolated center and surround stimuli. (D) Spike and excitatory synaptic input responses to the center-surround stimulus. Gray trace shows the linear sum of isolated responses to center- and surround-region stimuli. (E) Spike count in response to the center-surround stimulus compared to the linear sum of isolated center and surround responses. Each point is a different natural movie. Center and surround sum sub-linearly (On parasol RGCs: n = 20 natural movies across 8 cells, $p<9 \times 10^{-5}$; Off parasol RGCs: n = 18 natural movies across 7 cells, $p<2 \times 10^{-4}$). (F) Same as (E) but for excitatory charge transfer responses (On parasol RGCs: $p<2 \times 10^{-4}$; Off parasol RGCs: $p<3 \times 10^{-4}$). (G) For the example in (A–D), the difference between measured and linearly-summed spike responses was correlated with differences in excitatory synaptic inputs (r = 0.91). (H) Population data for the analysis in (G).

DOI: https://doi.org/10.7554/eLife.38841.015

simultaneous stimulation of both the center and surround regions (*Figure 6D*, black traces). For both spike and excitatory current responses, the measured center-surround response was smaller than the linear sum of the two responses measured independently. Thus, RF center and surround interact nonlinearly. This interaction, like that in *Figure 4 and 5*, is present in the excitatory synaptic input, and hence reflects properties of bipolar synaptic output rather than nonlinearities in synaptic integration or spike generation in the ganglion cell.

Sublinear interactions between center and surround inputs held across cells and fixations for both spike output (*Figure 6E*) and excitatory synaptic input (*Figure 6F*). For each cell, the difference between the linear sum of responses to center and surround inputs and the measured simultaneous response for spike outputs was correlated with the same difference for excitatory inputs (*Figure 6G, H*). This is consistent with the interpretation that the nonlinear interaction seen at the level of spike output is largely inherited from the excitatory synaptic inputs. Sublinear interactions in spike output and excitatory synaptic input were more strongly correlated for Off than On parasol RGCs (*Figure 6H*). This may be because inhibitory input impacts On parasol responses to natural stimuli more than Off parasol responses (*Turner and Rieke, 2016*). Taken together, these observations demonstrate that the nonlinear center-surround interactions characterized in *Figures 2–5* are prominent for naturalistic visual inputs.

## Natural spatial correlations promote nonlinear center-surround interactions

How do nonlinear center-surround interactions depend on stimulus statistics, especially those that characterize natural scenes? Naturalistic center and surround stimuli tended to elicit responses at different times (*Figure 6*). This is consistent with the spatial correlations in intensity that characterize natural images (*Simoncelli and Olshausen, 2001*) and the antagonistic nature of the surround— for example an Off parasol RGC would be depolarized by negative contrast in the RF center and hyperpolarized by negative contrast in the surround. We tested the effect of spatial correlations on nonlinear center-surround interactions using a synthetic visual stimulus inspired by our natural movie stimuli.

This stimulus consisted of a uniform disc in the center and a uniform annulus in the surround. The intensity of each region was sampled from a natural image (*Figure 7A,B*) and presented to either the center region alone, surround region alone, or both regions simultaneously. We updated the intensity of each region every 200 ms, which is consistent with typical human fixation periods (*Van Der Linde et al., 2009*). Center and surround intensities were determined from the mean intensity within the disc and annulus for randomly chosen image locations. The intensity correlations characteristic of natural scenes were evident when we plotted the center intensity against the corresponding surround intensity (*Figure 7C*, left, 'Control'). Shuffling the surround intensities relative to those of the center eliminated spatial correlations, while maintaining the same marginal distributions (*Figure 7C*, right, 'Shuffled'). When spatial correlations were intact, inputs to the center and surround combined sub-linearly in the excitatory synaptic input to the cell (*Figure 7D*), as they did in the full natural movie responses (*Figure 6*). When we shuffled the surround intensities relative to the center, nonlinear center-surround interactions were much weaker (*Figure 7E,F*). This was true for excitatory synaptic inputs to both On and Off parasol RGCs (*Figure 7G*).

To further probe the impact of intensity correlations on center-surround interactions, we generated Gaussian random noise stimuli that updated with a 200-ms period. For this stimulus, a single random intensity fills the entire center disc and a different, random intensity fills the entire surround annulus. This noise stimulus had a tunable degree of correlation between center and surround intensity, ranging from −1 (perfectly anti-correlated) to 0 (uncorrelated) to +1 (perfectly correlated, that is modulated in unison). When noise stimuli in the center and surround were negatively correlated, inputs to the center and surround summed linearly or very nearly so (*Figure 7H*, left). As the center-surround intensity correlations increased, sublinear interactions became more obvious (*Figure 7H*, middle and right). Strongly positively correlated noise stimuli induced center-surround interactions that resembled those seen using naturally correlated luminance stimuli (*Figure 7H*, right). This dependence of nonlinear center-surround interactions on center-surround intensity correlations was present in both On and Off parasol RGCs (*Figure 7I*).

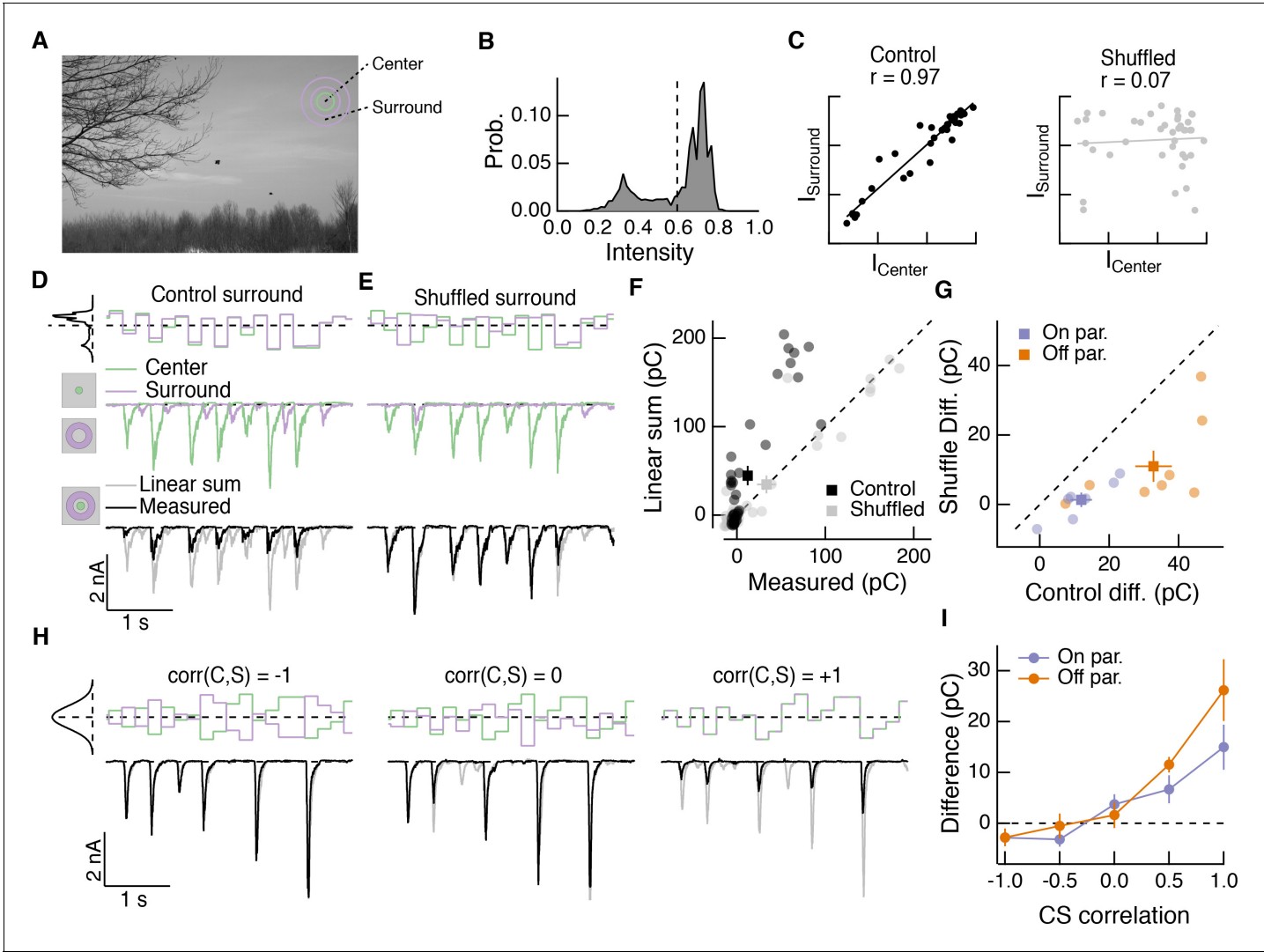

**Figure 7.** Spatial correlations in natural scenes promote nonlinear center-surround interactions. (**A**) Example image (*van Hateren and van der Schaaf, 1998*) used to construct natural intensity stimuli. (**B**) Intensity histogram from the image in (**A**). Dashed vertical line indicates the mean intensity, which was used as the mean gray level in experiments that follow. (**C**) Center and surround intensity values for 40 image patches from the image in (**A**). (**D**) Example stimuli (top) and Off parasol RGC excitatory current responses to isolated center and surround (middle) and center-surround (bottom) stimulation. Gray trace in bottom shows linear sum of isolated center and surround responses. (**E**) Same as (**D**) for shuffled surround intensities. (**F**) The response magnitude (charge transfer) of each fixation is plotted for measured center-surround and linearly summed center and surround responses. Circles show mean responses for each fixation, squares show mean (± S.E.M.) across all fixations in this example cell. (**G**) Population data showing the mean difference between responses to the center-surround stimulus and the linearly summed response. Circles show average differences for each cell tested, and squares show population mean (± S.E.M) (n = 7 On parasol RGCs, *p*<0.02; n = 8 Off parasol RGCs, *p*<8 × 10$^{-3}$). (**H**) White noise center-surround stimuli had variable center-surround correlations but constant marginal distributions. Shown are example excitatory current responses in an Off parasol RGC. Black traces show the measured center-surround stimulus response and gray traces show the linear sum of center and surround responses. (**I**) Population data from the experiments in (**H**) showing that nonlinear center-surround interactions depend on the correlation between center and surround inputs (n = 8 On parasol RGCs; n = 8 Off parasol RGCs).

DOI: https://doi.org/10.7554/eLife.38841.016

The following source data is available for figure 7:

**Source data 1.** Included is a .mat file containing a data structure for the data in *Figure 7A–G*.
DOI: https://doi.org/10.7554/eLife.38841.017

These results indicate that nonlinear center-surround interactions depend strongly on center-surround intensity correlations, and hence that the importance of these interactions could be underestimated from stimuli such as spatial noise that lack intensity correlations.

## A luminance-matched surround promotes spatial contrast sensitivity in the center

The experiments described above show that surround signals, rather than only interacting with the fully formed center signal, can alter how the center integrates over space. Two aspects of these results deserve emphasis: (1) nonlinear spatial integration is maximized when center and surround experience similar mean luminances (*Figures 2* and *3*) and (2) natural stimuli elicit strong nonlinear center-surround interactions due to positive correlations between center and surround intensities (*Figures 6* and *7*). These observations lead to the hypothesis, tested below, that surround activation can make spatial integration in the RF center relatively insensitive to changes in mean luminance.

Natural visual stimuli, such as the change in input encountered after a saccade, typically include changes in mean luminance and spatial contrast. Such stimuli will activate both linear and nonlinear response components, and these may not interact in a straightforward manner. To make this more concrete, consider a population of Off bipolar cells acting as subunits in the center of the RGC RF. At rest, the synapse of each bipolar cell is in a sharply rectified state. A decrease in mean luminance over the RF center will depolarize the synapse associated with each RF subunit, shifting each into a locally linear state (*Figure 8A*, top), and decreasing spatial contrast sensitivity. When such a stimulus is paired with a luminance-matched surround stimulus, the antagonistic surround may at least partially cancel this depolarization; this could keep each subunit in a locally rectified state (*Figure 8A*, bottom, blue arrow) and preserve spatial contrast sensitivity.

We tested this prediction in Off parasol RGCs, using modified grating stimuli with nonzero mean luminance. For each grating stimulus, we also presented a corresponding linear equivalent disc stimulus, which has the same mean luminance as the grating, but lacks any spatial contrast. The degree to which a cell's response to these two stimuli differs is a measure of the sensitivity of the cell to spatial contrast, or, equivalently, the strength of nonlinear spatial integration.

We presented these stimuli to Off parasol RGCs while measuring spike responses. A grating with a dark mean luminance signal produced a similar response as a linear equivalent stimulus (*Figure 8B*, left), indicating that the cell is insensitive to the spatial contrast present in the grating. Compare this to these cells' highly nonlinear responses to zero-mean grating stimuli (e.g. *Figure 3*). When a dark mean grating is paired with a luminance-matched surround, however, the grating again produces a much stronger response than its linear equivalent stimulus (*Figure 8B*, middle), which is consistent with the restoration of spatial contrast sensitivity via the mechanism in *Figure 8A*. Pairing the grating stimulus with poorly matched surrounds does not restore spatial contrast sensitivity (*Figure 8B*, right), indicating that this is not a general consequence of surround activation.

To quantify the spatial contrast sensitivity in these experiments, we used a nonlinearity index (NLI, See *Equation 1* and [*Turner and Rieke, 2016*]):

$$NLI = \frac{r_{image} - r_{disc}}{r_{image} + r_{disc}} \tag{1}$$

This measure normalizes responses within each surround condition. A positive NLI indicates that the cell responds more strongly to a grating stimulus than to its linear equivalent disc stimulus, and is thus sensitive to spatial contrast. A NLI near zero indicates that the cell's response is mostly determined by the mean luminance component of the stimulus, and not the spatial contrast. The NLI was maximal for all surround conditions for zero-mean gratings. As the center intensity decreased (moving towards the left in *Figure 8C*), the NLI decreased along with it, but this drop was less pronounced under the matched surround condition. Hence matched surround activation decreased the sensitivity of nonlinear spatial integration to changes in mean luminance.

## Intensity correlations in natural images promote nonlinear spatial integration

The results presented thus far show that systematically varying the input to the surround relative to the center can alter sensitivity to spatial contrast in both artificial and natural stimuli (*Figures 2*,

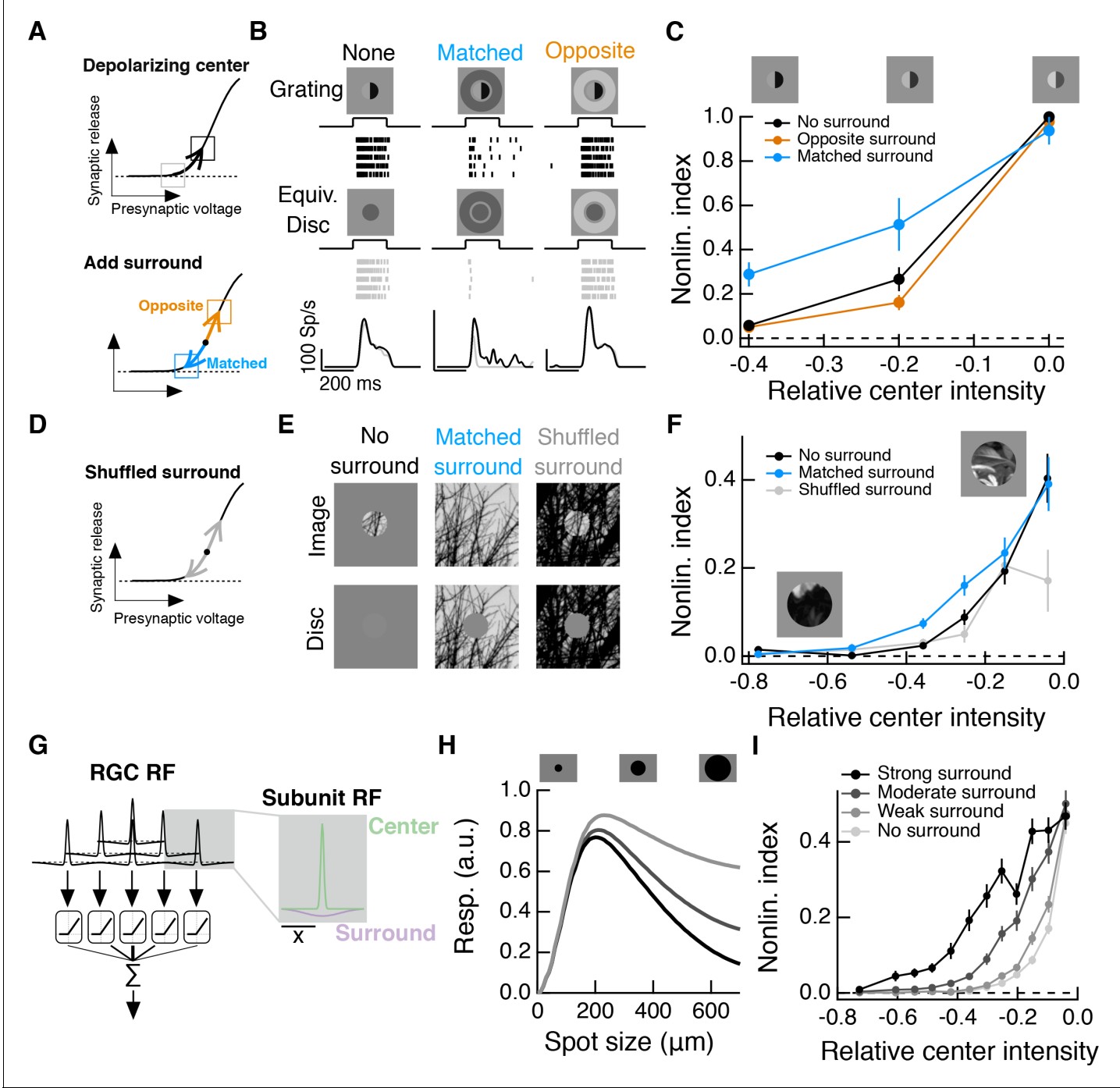

**Figure 8.** Intensity correlations across space promote nonlinear spatial integration in the RF center. (**A**) Schematic showing the hypothesized interaction between center and surround inputs on local subunit rectification. A depolarizing input to the center may push the synapse into a locally linear state. A simultaneous surround input that is matched in luminance (blue arrow) can hyperpolarize the synaptic terminal and bring the synapse back into a rectified state, whereas a poorly matched surround will not (orange arrow). (**B**) During Off parasol spike recordings, we presented split-field grating stimuli to the RF center under three surround conditions. For each stimulus condition, rows of the raster correspond to repeated presentations of the same stimulus for the example cell. (**C**) Summary data showing the population mean ± S.E.M. NLI (see text) as a function of the mean intensity (relative to the background) of the center grating (n = 8 Off parasol RGCs). (**D–F**) We presented natural image patches and their linear equivalent disc stimuli to measure the NLI under three surround conditions: no surround, a matched surround image, and a shuffled surround image. (**G**) Schematic of a nonlinear subunit RF model. Each subunit has a difference-of-Gaussians spatial receptive field. The output of each subunit is passed through a private, rectifying output nonlinearity. Subunit outputs are then summed over visual space to yield the modeled RGC response. (**H,I**) We changed the strength of the subunit surround to model RGCs with three different surround strengths: a weak surround (light gray trace), an intermediate-strength surround

*Figure 8 continued on next page*

Figure 8 continued

(gray trace), and a strong surround (black trace). We presented this RF model with the natural image/disc stimuli shown in (E) and, following that analysis, measured the NLI as a function of the mean intensity of the image in the RF center.

DOI: https://doi.org/10.7554/eLife.38841.018

The following figure supplements are available for figure 8:

**Figure supplement 1.** Center-surround intensity differences modulate spatial contrast sensitivity for randomly shuffled natural surrounds.

DOI: https://doi.org/10.7554/eLife.38841.019

**Figure supplement 2.** A spatiotemporal RF model also shows modulation of spatial contrast sensitivity with surround activation.

DOI: https://doi.org/10.7554/eLife.38841.020

*3* and *8A–C*). However, it is not clear from these experiments how much this effect is present during the course of more naturalistic activation of the RF surround. The intensity correlations present in natural images (e.g. *Figure 7C*) should ensure that the mean intensity difference between the RF center and surround is often near zero. Because spatial contrast sensitivity was maximized by small differences between mean center and surround intensity (*Figure 2*), we hypothesized that full natural image stimulation of the RF surround would increase spatial contrast sensitivity in the RF center compared to stimulation of the RF center alone.

To test this hypothesis, we presented natural image patches (and their corresponding linear equivalent disc stimuli) to the RF center while pairing each with three distinct surround conditions: no surround stimulation (*Figure 8E*); the naturally occurring surround present in the rest of the natural image patch ('Matched surround'), and, a randomly selected surround from the same full natural scene ('Shuffled surround'). We recorded Off parasol spike responses to these six stimuli for each of 20–40 randomly selected image patches from a single natural scene. For each image patch, we computed the NLI (*Equation 1*) for responses measured in each of the three surround conditions. We compared the NLIs to the mean intensity of the image in the RF center (I) relative to the background intensity (B), that is Relative center intensity $= (I - B)/B$. As in the modified gratings experiments (*Figure 8C*), a darker mean luminance signal was associated with a decrease in spatial contrast sensitivity (*Figure 8F*, black curve). This drop-off in spatial contrast sensitivity was less pronounced with the naturally occurring surround, (*Figure 8F*, blue curve). Specifically, sensitivity to fine spatial structure was two to three times greater in the presence of a matched surround than without a surround. Randomly selected surrounds did not enhance spatial contrast sensitivity in this way but instead altered the NLI in a manner predicted by the experiments in *Figure 2* (see *Figure 8—figure supplement 1*).

These observations are consistent with the idea that natural images provide inputs to the surround that can preserve the spatial contrast sensitivity of the RF center compared to center inputs alone. Key to this relative invariance of contrast sensitivity is the ability of the surround to control the degree of rectification of the bipolar subunits that comprise the RF center and the strong positive correlations between center and surround inputs created by natural images.

The appropriate surround activation can preserve spatial contrast sensitivity in the context of both natural image and grating stimuli. Note, however, that the NLI is, on average, lower for randomly selected images than for grating stimuli (compare *Figure 8F and C*). This is expected because the spatial structure of grating stimuli is designed to highlight nonlinear spatial integration by differentially activating subunits in the RF center (i.e. depolarizing some while hyperpolarizing others). Randomly-selected image patches, however, often do not contain much spatial structure that will differentially activate subunits in the RF center.

To explore the relationship between naturalistic surround activation and spatial contrast sensitivity in a manner not possible in our experiments, we constructed a simple spatial RF model composed of nonlinear, center-surround subunits (*Figure 8G*; see (*Enroth-Cugell and Freeman, 1987*) and Materials and methods). Following the analysis used for the data in *Figure 8F*, we computed the mean NLI for the model as a function of the relative center intensity for a surround-free stimulus (*Figure 8I*, 'no surround') and for the naturally occurring surround stimulus (*Figure 8I*, 'moderate surround'). As in the Off parasol spike data, the inclusion of the naturally occurring surround extended spatial contrast sensitivity in the face of stronger local luminance signals. We repeated the same analysis for versions of the RF model with both a weaker and a stronger surround. A stronger RF surround is associated with greater spatial contrast sensitivity, especially for images that contain a

strong local luminance signal. Similar results were seen for a spatiotemporal RF model that includes temporal filters measured in the experiments shown in *Figure 4* (*Figure 8—figure supplement 2*).

## Discussion

Here, we have shown that visual inputs to the RF surround can change how the RF center integrates inputs across space, including in natural scenes. During natural vision, the local luminance experienced by a RGC can vary dramatically from fixation to fixation (*van Hateren et al., 2002*). Because sensitivity to small scale (sub-RF center) spatial contrast relies critically on the rectification of bipolar subunit outputs, these changes in luminance and corresponding changes in bipolar voltage could prevent a RGC that responds nonlinearly to grating stimuli from detecting spatial contrast present in natural images. The incorporation of an antagonistic surround before formation of the nonlinear center ensures that this effect of the mean intensity signal is partially mitigated for natural scenes and other stimuli with strong spatial correlations in intensity. This mechanism is in line with the surround's classical role in reducing sensitivity to changes in mean luminance via subtraction, but it has the additional consequence of reducing the dependence of spatial contrast sensitivity on mean luminance by controlling the set point of the bipolar cell nonlinearity. The extent to which spatial contrast sensitivity is preserved will depend on the strength of the surround, and hence both cell type and properties of the visual environment like ambient illumination that can alter surround strength (*Barlow et al., 1957*). For those (relatively rare) regions of an image where the center and surround luminance signals are very different, a nonlinear RGC will lose much of its small-scale spatial contrast sensitivity and instead encode input with a spatial scale dictated by the classical center-surround RF. This could happen in a region of a scene that contains structure like an edge or a boundary between two large objects.

### Consequences for receptive field models

Past work in parasol RGCs as well as other RGC types has shown that the inclusion of nonlinear subunits in the RF center is important to capture responses to spatially structured stimuli (*Bölinger and Gollisch, 2012*; *Crook et al., 2008*; *Demb et al., 1999*; *Freeman et al., 2015*; *Schwartz et al., 2012*), including natural scenes (*Turner and Rieke, 2016*). The results shown here rely on nonlinear subunits in the RF center having a center-surround RF organization. This is consistent with previous observations that the surround is present in diffuse bipolar cells in primate retina (*Dacey et al., 2000*), and with past models of cat Y-type RGCs (*Enroth-Cugell and Freeman, 1987*). The work presented here indicates that capturing RGC center-surround interactions and spatial contrast sensitivity will require incorporating the impact of the surround on subunits within the receptive field center.

For the sake of concreteness, and based on past work in mouse retina (*Grimes et al., 2014*), we have assumed that the nonlinear relationship between bipolar cell input and excitatory input to the RGC arises at the bipolar cell to RGC synapse, but a contribution from nonlinearities earlier in the bipolar cell is likely as well. For example, active conductances in the axons of diffuse bipolar cells could shape the nonlinear relationship between bipolar cell input and synaptic release (*Puthussery et al., 2013*).

Importantly, we chose to focus on how systematically changing the mean intensity in the surround impacts coding in the RF center. This neglects the impact of spatial structure in the surround. Past work shows that RGC RF surrounds can show nonlinear spatial integration when probed with gratings and similar stimuli (*Takeshita and Gollisch, 2014*; *Demb et al., 1999*). If nonlinear spatial integration substantially shapes surround suppression, then the spatial structure of the surround should contribute to center-surround interactions. Indeed, in the salamander retina, Takeshita and Gollisch found that surround suppression was maximal when surround stimuli had a spatial scale several times larger than bipolar cell RFs (*Takeshita and Gollisch, 2014*), suggesting a role for larger amacrine or horizontal-cell-mediated subunits in the surround. This leads to the interesting prediction that spatial contrast sensitivity in the center will be enhanced when the surround contains spatial structure at a scale considerably larger than the fine structure encoded by the nonlinear center. However, for most RGC types, including parasol RGCs, little is known about the circuit origin or functional importance of nonlinear subunits in the RF surround (an exception is [*Demb et al., 1999*]). Understanding the nonlinear surround will likely be an important step toward building receptive field models that can account for responses to spatially structured stimuli like gratings and natural images.

The LN cascade modeling analysis in *Figure 4* allowed us to compare different forms of center-surround integration, but these models do not capture all the relevant mechanisms shaping a RGC's excitatory synaptic input. For example, history-dependent mechanisms like contrast and luminance adaptation probably substantially shape responses in a manner that these simple LN models cannot account for (*Demb, 2008*; *Howlett et al., 2017*). Other work has explored predictive models that can account for history-dependent mechanisms and thereby out-perform simpler LN models (*Howlett et al., 2017*; *Cui et al., 2016*; *Ozuysal and Baccus, 2012*). Future work could combine the architecture suggested by models that capture center-surround interactions (*Figure 4*, see also [*Enroth-Cugell and Freeman, 1987*]) with mechanisms to account for history-dependent gain changes.

## Relevance to other neural circuits

We focused here on Off parasol RGCs because of the strong nonlinear spatial integration seen in these cells, and its impact on the encoding of natural scenes (*Turner and Rieke, 2016*). We also showed that similar center-surround interactions occur in the excitatory synaptic inputs to On parasol RGCs. Indeed, any RGC that has rectified bipolar subunits with surrounds should show similar center-surround interactions and a similar effect on spatial contrast encoding, at least at the level of excitatory synaptic inputs. Direct inhibitory synaptic inputs to a RGC could further shape spatial integration and center-surround interactions; indeed, unlike Off parasol RGCs, the spike responses of On parasol RGCs to natural images are strongly shaped by direct inhibition (*Figure 6*, [*Turner and Rieke, 2016*]). An important topic for future work will be to determine how direct inhibitory synaptic input and center-surround interactions together shape spatial coding in On parasol cells and other RGC types.

Similar RF subregion interactions are present in other visual neurons. For example, activation of suppressive subunits in V1 RFs can change the shape of the nonlinear relationship between excitatory subunit activation and spike response (*Rust et al., 2005*). V1 surrounds are recruited when inputs to the surround are similar to those in the center (*Coen-Cagli et al., 2015*). The result is that the V1 surround is strongest in homogeneous visual contexts. Here, we see a similar contextual effect of retinal surrounds, albeit with a sensitivity to lower level statistical features. The surround has the greatest impact on RGC responses when visual stimuli contain luminance correlations (*Figures 6* and *7*).

The center-surround interactions observed here hint at a more general form of response modulation via parallel processing. Indeed, the circuit mechanisms that give rise to the observed center-surround interactions and modulation of spatial contrast sensitivity, namely nonlinear synaptic transfer functions and convergence of parallel neural pathways, are ubiquitous in neural circuits. The key circuit feature that creates unexpected interactions between parallel circuits is convergence prior to important nonlinear circuit elements—for example the bipolar output synapse. Such early convergence can cause one parallel circuit to regulate the effective nonlinearity applied to signals in the other circuit, and in doing so change the nature of the computation implemented by the circuit. Convergence could occur through rapid presynaptic inhibition, gap junctions, or integration of excitatory and inhibitory dendritic inputs in the presynaptic cell.

This work supports the broader notion that complex neural computations need not rely on exotic circuitry. Rather, interactions of known circuit elements can produce a variety of computations under the right stimulus conditions by producing modest changes in signaling that enhance or mitigate the impact of nonlinear mechanisms such as the bipolar output synapse. This highlights the importance of exploring how known neural circuit mechanisms interact under diverse stimulus conditions, including during naturalistic stimulation.

## Materials and methods

### Tissue preparation

We obtained retinal tissue from Macaque monkeys (*M. nemestrina*, *M. mulatta*, or *M. fascicularis*) via the tissue distribution program at the Washington National Primate Research Center. All procedures were approved by the Institutional Animal Care and Use Committee at the University of Washington. Dissection procedures have been described previously (*Angueyra and Rieke, 2013*; *Turner and*

*Rieke, 2016*). After enucleation, the eye was hemisected and the vitreous humor was removed mechanically, sometimes assisted by treatment with human plasmin ($\sim$50 $\mu$g/mL, Sigma or Haematologic Technologies Inc.). Retina was dark adapted for $\sim$1 hr, and all subsequent procedures were performed under infrared light using night-vision goggles. The retina and pigment epithelium were separated from the sclera and stored in oxygenated (95%$O_2$/5%$CO_2$) Ames bicarbonate solution (Sigma) in a light-tight container. Retinal mounts were removed from the pigment epithelium and laid photoreceptor-side down onto a poly-D-lysine coated coverslip (BD biosciences). All recordings were made from peripheral retina, at eccentricities exceeding 30˚. During experiments, retinal tissue was perfused at 7–9 mL/min with Ames solution at $\sim$32°C.

## Patch recordings

Electrophysiological recordings were performed using a Multiclamp 700B amplifier (Molecular Devices). Spike responses were measured using extracellular or loose-patch recordings with an Ames-filled pipette. For voltage clamp recordings, we used low-resistance pipettes (tip resistance $\sim$1.5–4 M$\Omega$) filled with a Cs-based internal solution (containing, in mM: 105 CsCH3SO3, 10 TEA-Cl, 20 HEPES, 10 EGTA, 5 Mg-ATP, 0.5 Tris-GTP, and 2 QX-314, pH 7.3, $\sim$280 mOsm). We compensated for access resistance ($\sim$4–8 M$\Omega$) online by 75%. Reported voltages have been corrected for an approximately $-10$ mV liquid junction potential. To measure excitatory synaptic inputs in voltage clamp recordings, we held the cell at the expected reversal potential for inhibitory inputs. This was typically around $-60$ mV, but was adjusted for each cell by delivering light steps at holding potentials near this value until the inhibitory response was eliminated.

## Cell identification and selection

We identified On and Off parasol RGCs under infrared illumination based on soma size and morphology as well as characteristic spike responses to light steps centered on the cell. The overall health and sensitivity of the retina was confirmed by delivering a uniform, 5% contrast, 4 Hz modulated stimulus, which produces a robust spike response (>40 spikes per second) in On parasol RGCs in sufficiently sensitive tissue. Sensitivity was continuously monitored (typically before each recording) in this way.

## Visual stimulation

Stimuli were presented and data acquired using custom written stimulation and acquisition software packages Stage (stage-vss.github.io) and Symphony (symphony-das.github.io). Lab-wide acquisition packages can be found at https://github.com/Rieke-Lab/riekelab-package (*Cafaro, 2018*; copy archived at https://github.com/elifesciences-publications/riekelab-package) and protocols used in this study can be found at https://github.com/Rieke-Lab/turner-package (*Turner, 2018c*; copy archived at https://github.com/elifesciences-publications/turner-package).

Visual stimuli were presented with 60 Hz frame rates on an OLED microdisplay monitor (eMagin, Bellevue, WA) focused onto the photoreceptors. Monitor outputs were linearized by gamma correction. Stimuli were calibrated using monitor power outputs, the spectral content of the monitor, macaque photoreceptor spectral sensitivity (*Baylor et al., 1987*), and a collecting area of 0.37 $\mu m^2$ for cones (*Schnapf et al., 1990*) and 1 $\mu m^2$ for rods. Unless otherwise noted, mean light levels produced $\sim$9,000 isomerizations (R\*)/M or L-cone/s, $\sim$2,000 R\*/S-cone/s and $\sim$18,000 R\*/rod/s.

Before every parasol RGC recording, we found the center of the cell's RF using a split-field contrast reversing grating stimulus at 4 Hz and 90% contrast. To do this, we translated the grating until the two F2 response cycles were balanced (i.e. we minimized the F1 while maximizing the F2 component of the response). We performed this search in both the horizontal and vertical dimensions. In experiments, where we specifically targeted visual stimuli to the RF center or surround, we measured each cell's area-summation curve and fit, online, to these data a circular difference-of-Gaussians RF model in radial coordinates. This model is described in *Equation 2* below.

$$R = K_C \times \left( 1 - exp\left( -\frac{r^2}{2\sigma_C^2} \right) \right) - K_S \times \left( 1 - exp\left( -\frac{r^2}{2\sigma_S^2} \right) \right) \qquad (2)$$

where $R$ is the response of the cell, $r$ is the spot radius, and four free parameters describe the shape of the RF, two for each center and surround: $K$ describes the amplitude scaling of each component

and $\sigma$ describes the size. We then chose a boundary for the center that would minimally activate the surround while still filling much of the center and likewise for the surround (e.g. see *Figure 2—figure supplement 1*). These boundaries were used to generate appropriate masks and apertures to target each RF subregion.

## Natural visual stimuli

Naturalistic movie stimuli (*Figure 6*) were generated using data from the DOVES database (*Van Der Linde et al., 2009*) http://live.ece.utexas.edu/research/doves/. The images in the DOVES database were selected from the van Hateren natural image database (*van Hateren and van der Schaaf, 1998*), and we used the original van Hateren images instead of the images included in the DOVES database. This is because the DOVES images are cropped, and using the original, larger images allowed for a wider selection of eye movement trajectories to be used. To select data from the DOVES database to use in experiments, we ensured that the eye trajectory never extended close enough to the boundaries of the image that our presented frames would extend beyond the boundaries of the image.

For natural movie (*Figure 6*), luminance (*Figure 7*), and image stimuli (*Figures 2* and *8*), we scaled each image such that the brightest pixel in the image was assigned an intensity value of 1 (maximum monitor intensity). The mean gray level of the monitor was set to the mean pixel intensity over the entire image. This mean gray level was used in masked/apertured regions of the frame as well as the blank screen between trials. Natural movies were presented at a spatial scale of 1 arc-min/pixel, which is equal to 3.3 $\mu m$/pixel on the monkey retina. Natural images, from the van Hateren database, were presented at a scale of 6.6 $\mu m$/pixel on the retina.

In the experiments in *Figure 7*, we updated the intensity of a disc (annulus) in the center (surround) every 200 ms, which is consistent with typical human fixation periods during free-viewing (*Van Der Linde et al., 2009*), although less than the mean period (for efficient data collection). To compute natural intensity stimuli for the center and surround, we selected many random patches from a natural image and measured the mean intensity within a circle of diameter 200 $\mu m$ (for the RF center) and within an annulus with inner and outer diameters 200 and 600 $\mu m$, respectively (for the RF surround).

For the natural image experiments in *Figure 2*, we used a natural image patch selection method that has been described previously (*Turner and Rieke, 2016*). Briefly, we used a nonlinear subunit model developed in *Turner and Rieke (2016)* to rank image patches based on their degree of response nonlinearity (i.e. how differently they drove a spatially linear compared to a spatially nonlinear RF model). We then sub-sampled the entire distribution of image patches in order to present an approximately uniform distribution of images that ranged from very nonlinear (i.e. high spatial contrast) to very linear (i.e. low spatial contrast). This ensured that we could efficiently explore the full range of spatial contrasts present in natural images within a single experiment. For the natural image experiments and modeling in *Figure 8*, we did not use this biased sampling method, instead selecting image patches randomly from within a natural image.

## Data analysis and modeling

Data analysis was performed using custom written scripts in MATLAB (Mathworks). The code used to analyze the data in this study, perform the computational modeling, and generate the figures presented can be found at https://github.com/mhturner/CenterSurroundInteractions (*Turner, 2018a*; copy archived at https://github.com/elifesciences-publications/CenterSurroundInteractions), and more general analysis code used in this study can be found in https://github.com/mhturner/MHT-analysis (*Turner, 2018b*; copy archived at https://github.com/elifesciences-publications/MHT-analysis). Throughout the study, reported p-values were computed using a Wilcoxon signed-rank test.

The center-surround models in *Figure 4* were constructed as linear-nonlinear cascade models. Using trials where only the center or surround was stimulated in isolation, we used reverse correlation between the cell's response and the noise stimulus to compute the linear filter for both the center and the surround. Each of the three models presented in *Figure 4* took as inputs the filtered center and surround noise stimuli.

For each of the nonlinearities in the models, we parameterized a smooth curve based on a cumulative Gaussian function to fit to the mean response as a function of generator signal (see [*Chichilnisky, 2001*]). Here, $C()$ is the cumulative normal distribution (*normcdf* in MATLAB).

$$N(x) = \alpha \times C(\beta x + \gamma) \tag{3}$$

*Equation 3* above shows the general form of this nonlinear function, which is determined by three free parameters: $\gamma$ is an offset along the horizontal axis, $\beta$ determines the sensitivity or slope of the contrast response function, and $\alpha$ is a scale factor which determines the maximal response. Equations for the three models are described in more detail below.

(1) The **Independent model** is described by seven free parameters. Three each for each of the independent nonlinearities of the form described in *Equation (3)* above, and one, $\epsilon$, which sets the vertical offset of the output response. In this and the following equations, $h_c$ ($h_s$) is the linear filter for the RF center (surround), $s_c$ ($s_s$) is the stimulus in the center (surround) and the * operator indicates a linear convolution.

$$R = \epsilon + N_c(h_c \star s_c) + N_s(h_s \star s_s) \tag{4}$$

(2) The **Shared model** is described by five free parameters. Three for the single nonlinearity (*Equation 3*), a vertical offset of the response ($\epsilon$), and a scale factor, $a$, which sets the relative weight of the center compared to the surround.

$$R = \epsilon + N(a \times (h_c \star s_c) + h_s \star s_s) \tag{5}$$

(3) The **Stacked model** is described by ten free parameters. Three for each of the three nonlinearities (*Equation 3*) and a single vertical offset, $\epsilon$.

$$R = \epsilon + N_{shared}(N_c(h_c \star s_c) + N_s(h_s \star s_s)) \tag{6}$$

To fit and test these models, we collected ~30 trials total, 10 each of center, surround, and center-surround conditions. We tested the models' predictive power against a single center-surround trial, fitting the model with the remaining trials. We repeated this for every trial, holding each out from the model fitting for testing purposes. The reported r-squared values are the average over these fitting iterations. Because we tested the models using single-trial responses (rather than the mean response to repeated presentations of frozen noise), some response variance reflects trial-to-trial variability in the retinal response and is not expected to be fit by these models. To estimate the retinal reliability, which sets an upper bound on the performance of these models, we used a subset of cells (n = 2 On parasol RGCs, n = 3 Off parasol RGCs) for which we did record responses to repeated presentations of frozen noise. Using these data, we computed the r-squared value for each single trial relative to the mean responses over all trials, and averaged across single trials to yield a single r-squared value. Using these estimates of the response reliability, the average fraction of 'explainable' variance was 0.81 and 0.75 for On and Off parasol RGCs, respectively.

For the image computable model in *Figure 8*, we converted image patches to Weber contrast ($C$) by normalizing each pixel intensity ($I$) to the mean over the entire natural image ($B$), that is,

$$C = \frac{I - B}{B} \tag{7}$$

The models in *Figure 8* and *Figure 8—figure supplement 2* are of an Off-center RF, so contrast-converted images were multiplied by $-1$. For both expanding spots stimuli and natural images in the spatiotemporal model, stimuli were turned on for 200 msec from a mean-gray level, as in the experiments. To compute the spatial activation of each subunit, we first convolved an image stimulus with one of two spatial filters: a Gaussian representing the subunit center, or a Gaussian representing the subunit surround. Our aim was to investigate how different RF architectures impacted center-surround interactions. Hence, the parameters describing the RF sizes were not fit to individual cells, but instead selected to be representative of RF center, surround, and subunit sizes typical of Off parasol RGCs at the retinal eccentricities used for these experiments. Our conclusions were not sensitive to exact RF size. Subunits were placed on a square grid in visual space. Each subunit's RF sub-field activation ($a_c$ for center activation, $a_s$ for surround activation) was computed by sampling

the appropriate convolved image at that subunit's location. For the spatiotemporal model in **Figure 8—figure supplement 2**, the 'voltage' signal of each subunit ($V_i$, i.e. the pre-nonlinearity activation), as a function of time is given by **Equation 8**,

$$V_i(t) = a_{c,i} \times [h_c \star s(t)] - w_s a_{s,i} \times [h_s \star s(t)] \tag{8}$$

where $h_c$ and $h_s$ are the temporal linear filters for center and surround, respectively. The filters we used were Off parasol excitatory current population means from the center-surround white noise experiments in **Figure 4**. $w_s$ is the weight of the subunit surround, relative to the center, and $s(t)$ is the stimulus waveform (in this case, a 200 ms step from zero). Again, the * operator indicates a linear convolution. The final step in computing the spatiotemporal model's response is to treat each subunit voltage, $V_i(t)$, with a rectifying nonlinear function and sum over all subunit outputs. This is given by **Equation 9**,

$$R(t) = \sum_{i=1}^{n} w_i \times f(V_i(t)) \tag{9}$$

where $w_i$ is the weighting of each subunit, defined by a circular Gaussian function over the RF center; the nonlinear transform applied to each subunit activation, $f$, was a simple threshold-linear function. Examples of the time-varying model responses, $R(t)$, can be seen in **Figure 8—figure supplement 2B**. To compute the responses in **Figure 8—figure supplement 2C,D** we integrated time varying responses over the duration of the 200 ms step (as in the analyses used in our experiments).

For the simpler spatial RF model (with no temporal component) in **Figure 8**, the pre-nonlinearity voltage of each subunit and response of the model RF is given by **Equations 10 and 11** below,

$$V_i = a_{c,i} - w_s a_{s,i} \tag{10}$$

$$R = \sum_{i=1}^{n} w_i \times f(V_i) \tag{11}$$

The RF model is described by four fixed parameters: the size of the Gaussian subunit center, ($\sigma$=10 $\mu m$ for the results shown here), the size of the subunit surround ($\sigma$=150 $\mu m$), the size of the RF center, used to weight subunit outputs ($w_i$ in **Equation 9** above, $\sigma$ = 40 $\mu m$), and the strength of the subunit surround, relative to the strength of the subunit center (0.5 for the 'Weak surround' model, 1.0 for the 'Moderate surround,' and 1.5 for the 'Strong surround').

The rectification index (RI) used in **Figure 5** is given by **Equation 12** below,

$$RI = 1 - \frac{r_0 - r_-}{r_+ - r_0} \tag{12}$$

where $r_0$ is the mean conductance response to zero center activation, $r_-$ is the response to maximal center deactivation (i.e. negative center activation), and $r_+$ is the response to maximal center activation.

## Acknowledgements

We thank Shellee Cunnington and Mark Cafaro for excellent technical support. Tissue was provided by the Tissue Distribution Program at the Washington National Primate Research Center (WaNPRC), and we are grateful for assistance from the WaNPRC staff, especially Chris English and Drew May. Raunak Sinha and Mike Manookin assisted in tissue preparation. We thank Nora Brackbill, Jon Cafaro, Mike Manookin, Luis Gonzalo Sánchez Giraldo and Odelia Schwartz for helpful feedback on an earlier version of this manuscript. This work was supported by NIH grants F31-EY026288 (MHT) and EY028542 (FR) and the Howard Hughes Medical Institute (FR).

## Additional information

### Competing interests

Fred Rieke: Reviewing editor, *eLife*. The other authors declare that no competing interests exist.

### Funding

| Funder | Grant reference number | Author |
|---|---|---|
| National Eye Institute | F31-EY026288 | Maxwell H Turner |
| National Eye Institute | EY028542 | Fred Rieke |
| Howard Hughes Medical Institute | | Fred Rieke |

The funders had no role in study design, data collection and interpretation, or the decision to submit the work for publication.

### Author contributions

Maxwell H Turner, Conceptualization, Data curation, Formal analysis, Validation, Investigation, Visualization, Methodology, Writing—original draft, Writing—review and editing; Gregory W Schwartz, Conceptualization, Writing—review and editing; Fred Rieke, Conceptualization, Supervision, Funding acquisition, Investigation, Methodology, Writing—original draft, Project administration, Writing—review and editing

### Author ORCIDs

Maxwell H Turner (iD) http://orcid.org/0000-0002-4164-9995
Fred Rieke (iD) http://orcid.org/0000-0002-1052-2609

### Ethics

Animal experimentation: Tissue was obtained via the tissue distribution program at the Washington National Primate Research Center. All animal procedures were performed in accordance with IACUC protocols at the University of Washington (IACUC protocol number 4277-01).

### Decision letter and Author response

Decision letter https://doi.org/10.7554/eLife.38841.024
Author response https://doi.org/10.7554/eLife.38841.025

## Additional files

### Supplementary files

• Source code 1. Example MATLAB code to load the Source Data files, plot responses, and perform basic analyses. The analysis used to generate the figures in the paper is included in the corresponding GitHub repository (see Materials and methods) – this Source Code file is merely illustrative of how to access and interact with the provided Source Data files.
DOI: https://doi.org/10.7554/eLife.38841.021
• Transparent reporting form
DOI: https://doi.org/10.7554/eLife.38841.022

### Data availability

We have made all the data in the study freely available. Source data files have been provided for Figures 2, 3, 4 and 7, and example code to demonstrate how to pull out and plot the data is provided as Source code file 1.

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
