## [Decision Letter]

Thank you for submitting your article "Receptive field center-surround interactions mediate context-dependent spatial contrast encoding in the retina" for consideration by *eLife*. Your article has been reviewed by three peer reviewers, and the evaluation has been overseen by Andrew King as the Senior and Reviewing Editor. The reviewers have opted to remain anonymous.

The reviewers have discussed the reviews with one another and the Reviewing Editor has drafted this decision to help you prepare a revised submission.

Summary:

The antagonistic receptive fields of many retinal ganglion cells consist of two components, a center and a surround responding oppositely to light. Models of these receptive fields assume that the surround is added after the center response is established. However, evidence from many species, including mouse and primate retina, suggests that center-surround antagonism is established at the level of cones and bipolar cells through lateral inhibition from horizontal and amacrine cells, i.e. before retinal ganglion cells summate their input. The authors explored the functional consequences of this receptive field organization in OFF parasol retinal ganglion cells of the primate retina with patch-clamp recordings and modeling. Using sophisticated stimulus paradigms, they found that nonlinear spatial integration was maximal when center and surround were stimulated with a similar mean luminance – a correlation often present in natural scenes. These results show that activation of the surround switches the visual computation performed by the receptive field center from encoding information about local luminance to encoding information about spatial contrast in the visual scene.

The reviewers agreed that these experiments have been expertly performed, analyzed and presented. The findings are exciting and advance our understanding of how receptive field organization may enable retinal ganglion cells to maintain their contrast sensitivity independently of mean luminance. This paper may change the way we think about the center-surround organization of retinal ganglion cells and has implications for many other circuits in the CNS.

Essential revisions:

A question was raised about potential relevance to other circuits: The analysis seems to suggest that the presynaptic inhibition exhibits fast temporal dynamics that are required for the computations. This is certainly plausible in the retina, where bipolar cells receive presynaptic inhibition mediated by fast GABA-A and glycine receptors. In many brain circuits, however, presynaptic inhibition is mediated by slower metabotropic receptors. Is this kind of processing less likely to occur in those regions? Information can be processed on multiple time scales, of course, but this raises the question as to whether the presence of presynaptic ionotropic inhibitory receptors is a prerequisite for this kind of signal processing on a faster time scale. It would be useful to address this at the end of the Discussion.

Work from Tim Gollisch's lab has shown that surround integration varies with spatial scale. It would have been interesting to probe receptive fields with patches of natural scenes in which light intensity is maintained and the spatial scale of the surround varied (instead of shuffling surrounds and thereby changing light intensity/contrast). The authors should relate their findings to this previous work.

One of the reviewers felt that you should state at the beginning of the article that this study focuses on OFF parasol retinal ganglion cells. It was also suggested that you should mention OFF retinal ganglion cells in the title and perhaps transfer data on ON parasol retinal ganglion cells to the supplementary information. However, it was agreed to leave this up to you.

---

## [Author Response]

Essential revisions:A question was raised about potential relevance to other circuits: The analysis seems to suggest that the presynaptic inhibition exhibits fast temporal dynamics that are required for the computations. This is certainly plausible in the retina, where bipolar cells receive presynaptic inhibition mediated by fast GABA-A and glycine receptors. In many brain circuits, however, presynaptic inhibition is mediated by slower metabotropic receptors. Is this kind of processing less likely to occur in those regions? Information can be processed on multiple time scales, of course, but this raises the question as to whether the presence of presynaptic ionotropic inhibitory receptors is a prerequisite for this kind of signal processing on a faster time scale. It would be useful to address this at the end of the Discussion.

This is an interesting point – thank you for bringing it up. We have added a paragraph to the Discussion (under the subsection entitled “Relevance to other neural circuits”) that discusses this point. We agree that metabotropic presynaptic inhibition would be better suited for slow-timescale processing than for the dynamic changes in computation we observed in the parasol circuit. Presynaptic inhibition is also not a requisite feature for the form of center-surround interactions that we observe. For example, inhibitory synaptic input to the dendrites of a cell comprising a RF subunit could control the voltage and nonlinearity of the subunit output in much the same way as axonal (presynaptic) inhibitory input to the same cell. The key point is that the combination of signals occurs prior to the subunit nonlinearity. We now note this possibility in the Discussion as well.

Work from Tim Gollisch's lab has shown that surround integration varies with spatial scale. It would have been interesting to probe receptive fields with patches of natural scenes in which light intensity is maintained and the spatial scale of the surround varied (instead of shuffling surrounds and thereby changing light intensity/contrast). The authors should relate their findings to this previous work.

Thank you for pointing this out. We agree that this (and related work from Demb et al. (1999)) is an interesting and relevant finding. We added a paragraph to the Discussion (third paragraph in the subsection entitled “Consequences for receptive field models”) that relates our findings to this work. Studying natural images involves many decisions about which features to manipulate; we hope in future studies to build on the work here to more completely explore the full set of features of natural images that control surround activity; nonlinear spatial integration and sensitivity to spatial structure in the surround will certainly need to be included in a more complete understanding.

One of the reviewers felt that you should state at the beginning of the article that this study focuses on OFF parasol retinal ganglion cells. It was also suggested that you should mention OFF retinal ganglion cells in the title and perhaps transfer data on ON parasol retinal ganglion cells to the supplementary information. However, it was agreed to leave this up to you.

Thank you. We agree that the Introduction of the original manuscript suggested this study would focus on both On and Off parasol RGCs in equal measure, while the results did not follow through with that. However, we prefer to keep the On parasol RGC results in the main text, alongside the Off parasol data, for two reasons:

1) We feel that the presence of similar center-surround interactions in the On parasol excitatory inputs is interesting and suggests that these interactions might be a general feature of RGC RFs rather than a peculiarity of Off parasols. The impact on spike output is more complicated for On parasol RGCs because, in part, of strong inhibitory inputs that shape nonlinear responses in On parasols. This issue is beyond the scope of the current study, and we are currently finishing a study dealing with the impact of inhibitory inputs on spatial integration by On parasol RGCs. We have revised the text to point out these issues (see below).

2) For presentation purposes, it is much cleaner and simpler to present the On parasol data alongside the Off data in Figures 4-7, since the experiments and analyses are the same. This also facilitates interesting comparisons between On and Off cells, e.g. stronger rectification in Off compared to On parasol excitatory input (Figure 5), greater correlation between excitatory input and spike output in Off compared to On parasol RGCs (Figure 6), and the stronger nonlinear center-surround interactions in Off compared to On parasol RGCs (Figure 7).

That being said, we did not communicate the importance or relevance of the On parasol results clearly in the original version of the manuscript. We have revised the text in a few places to more accurately set reader expectations and communicate how the On parasol results are relevant. In particular, see:

1) The introduction, where we explicitly say that we focus on Off parasol RGCs.

2) The Results introducing Figure 2—figure supplement 1, where we now point out that the presence of similar center-surround interactions in On parasol excitatory inputs suggests a generality to this phenomenon, or at least that it is not a feature unique to Off parasols.

3) The Results introducing Figure 4 (subsection “Nonlinear center-surround interactions are dominated by a single, shared nonlinearity”).

4) The final paragraph of the subsection entitled “RF center and surround interact nonlinearly during naturalistic visual stimulation.”

5) The Discussion, first paragraph of subsection entitled “Relevance to other neural circuits.”

If the reviewers continue to feel that the On parasol data are distracting in the revised manuscript, we will move it to the supplement. We have also kept the title general, but are happy to reconsider if the reviewers feel that the title should indicate a focus on Off parasol cells.